# Comparative Genomic Analysis of *SAUR* Gene Family, Cloning and Functional Characterization of Two Genes (*PbrSAUR13* and *PbrSAUR52*) in *Pyrus bretschneideri*

**DOI:** 10.3390/ijms23137054

**Published:** 2022-06-24

**Authors:** Mengna Wang, Muhammad Aamir Manzoor, Xinya Wang, Xiaofeng Feng, Yu Zhao, Jinling He, Yongping Cai

**Affiliations:** School of Life Sciences, Anhui Agricultural University, Hefei 230036, China; wmn970706@163.com (M.W.); aamir.manzoor@stu.ahau.edu.cn (M.A.M.); wangxinya19971223@163.com (X.W.); 15858221779@163.com (X.F.); zy18756176283@163.com (Y.Z.); hjl@ahau.edu.cn (J.H.)

**Keywords:** *SAUR* gene, Dangshan pear, stone cell, lignin synthesis

## Abstract

The *SAUR* (small auxin-up RNA) gene family is the biggest family of early auxin response genes in higher plants and has been associated with the control of a variety of biological processes. Although *SAUR* genes had been identified in several genomes, no systematic analysis of the *SAUR* gene family has been reported in Chinese white pear. In this study, comparative and systematic genomic analysis has been performed in the *SAUR* gene family and identified a total of 116 genes from the Chinese white pear. A phylogeny analysis revealed that the *SAUR* family could be classified into four groups. Further analysis of gene structure (introns/exons) and conserved motifs showed that they are diverse functions and *SAUR*-specific domains. The most frequent mechanisms are whole-genome duplication (WGD) and dispersed duplication (DSD), both of which may be important in the growth of the *SAUR* gene family in Chinese white pear. Moreover, cis-acting elements of the *PbrSAUR* genes were found in promoter regions associated with the auxin-responsive elements that existed in most of the upstream sequences. Remarkably, the qRT-PCR and transcriptomic data indicated that *PbrSAUR13* and *PbrSAUR52* were significantly expressed in fruit ripening. Subsequently, subcellular localization experiments revealed that *PbrSAUR13* and *PbrSAUR52* were localized in the nucleus. Moreover, *PbrSAUR13* and *PbrSAUR52* were screened for functional verification, and Dangshan pear and frandi strawberry were transiently transformed. Finally, the effects of these two genes on stone cells and lignin were analyzed by phloroglucinol staining, Fourier infrared spectroscopy, and qRT-PCR. It was found that *PbrSAUR13* promoted the synthesis and accumulation of stone cells and lignin, *PbrSAUR52* inhibited the synthesis and accumulation of stone cells and lignin. In conclusion, these results indicate that *PbrSAUR13* and *PbrSAUR52* are predominantly responsible for lignin inhibit synthesis, which provides a basic mechanism for further study of *PbrSAUR* gene functions.

## 1. Introduction

Auxin (Indole acetic acid, IAA) is one of the earliest discovered plant hormones. It plays an important regulatory role in the process of plant growth and development. It can induce the expression of a series of auxin early response genes [1]. Small auxin up RNA (*SAUR*) is a gene that responds most rapidly and violently to auxin. They can respond to active auxin genes within 2 min~5 min and encode relatively special small molecular proteins in much of the vegetation [2,3]. The *SAUR* gene family is the largest family of plant-specific auxin response factors. *SAUR* genes generally are intronless genes, and most of them exist in clusters [4]. The relative molecular weight of the protein they encode is relatively small size between 9 × 10^3^ and 3 × 10^4^ and can be synthesized in a very short time after auxin treatment [4]. Due to the existence of a conserved downstream element (DST) in the 3′ untranslated region of *SAUR* [5,6,7], the mRNAs encoded by *SAUR* are extremely unstable and will be degraded within a few minutes. At the same time, similar to the other two types of auxin early response genes, most *SAUR* genes contain one or more auxin response elements (AuxRES) in the promoter region [8,9,10,11,12,13,14]. In addition, most SAUR proteins have a conservative *SAUR*-specific domain (SSD) with about 60 amino acid residues [15] For example, *Arabidopsis thaliana*, *Oryza sativa*, and *Zea mays* have a *SAUR* domain consisting of 60 amino acids.

China is the origin of the pear. It not only has rich pear variety resources but is also the largest pear-producing country in the world. The cultivation area and yield account for more than 70% of the world [16]. The content, size, number, and density of stone cell clusters in fruit affect the texture, taste, and flavor of fruit [17]. ‘Dangshan Su pear’ (*Pyrus bretschneideri* cv. *Dangshan Su*) is the pear variety with the largest cultivation area in China [18]. Its inherent defects of high content and large diameter of stone cell clusters have become increasingly prominent, seriously affecting its quality and processing. It has become an important theoretical and technical bottleneck problem to be solved in the pear industry in China to improve fruit quality by regulating the development of stone cell mass.

The stone cell is a kind of thick-walled cell with a supporting function, which is mainly composed of lignin and cellulose [19]. Lignin accounts for 18% and is an important part of stone cells [20]. According to the monomer composition of lignin polymer, lignin can be divided into three types: Guaiacyl lignin (G-type lignin), P-hydroxyphenyl lignin (H-type lignin), Syringyl lignin (S-type lignin) [21].

Auxin early response factor (*SAUR*) is a class of genes that respond most rapidly and violently to auxin, and auxin promotes the activity of cambium specialization and leads to the development of conductive tissue with high lignin content [22]. Therefore, it is speculated that *SAUR* gene may affect the accumulation of lignin. The transcriptomic data of ‘Dangshansu Pear’ fruit at different developmental stages were downloaded from NCBI, and it was found that *PbrSAUR13* and *PbrSAUR52* were significantly differentially expressed in the *PbrSAUR* gene family during fruit ripening. At present, it has not been reported yet that *SAUR* regulates the synthesis of pear fruit stone cells and lignin. Therefore, taking *PbrSAUR13* and *PbrSAUR52* key genes, and studying the possible role of *PbrSAUR* in regulating the lignification of pear stone cell clusters will not only help to clarify the molecular mechanism of lignification of pear stone cell clusters but also provide new ideas for improving the quality of pear fruit.

## 2. Results

### 2.1. Identification and Phylogenetic Analysis of SAUR Genes

A total of 116 *SAUR* proteins were identified and used for further analysis (Appendix A), of which there were 63 *SAUR* genes with isoelectric points (pIs) greater than 7, the lowest pI value was 5.1 (*Pbr003399.1* and *PbrSAUR5*), whereas the highest pI value was 10.28 (*PbrSAUR11*, and *PbrSAUR10*). The molecular weights of these *SAUR* genes vary greatly, with the minimum molecular weight of 7.47 kDa (*PbrSAUR54*) and the maximum molecular weight of 122.22 kDa (*PbrSAUR91*). The number of amino acids in the *SAUR* proteins were also varied widely, with *PbrSAUR21* containing the fewest amino acids (only 67), whereas, *PbrSAUR6* had the largest amino acid number (1090). The grand average hydrophilicity (GRAVY) of all 116 *SAUR* genes was less than 1, and the smallest GRAVY value (−0.811) was obtained for *PbrSAUR4*.

A phylogenetic tree of 239 *SAUR* genes including ‘Dangshansu pear’ (*Pyrus bretschneideri*), *Arabidopsis thaliana* and *Oryza sativa* was constructed using the ML method (Figure 1). According to the phylogenetic analysis, we identified four phylogenetic groups (groups 1–4). Group 1 had the most members, including 87 *SAUR* genes from ‘Dangshansu pear’ (*Pyrus bretschneideri*), *Arabidopsis thaliana* and *Oryza sativa*. Group 3 had the fewest members, with only 37 *SAUR* genes, and groups 2 and 4 contained 61 and 54 *SAUR* genes, respectively. *PbrSAUR* gene family contribute to all group with *Arabidopsis thaliana* and *Oryza sativa.* These findings showed that these were gene loss or gain events that happened during the evolutionary process. The addition and deletion of certain *SAUR* gene members resulted in functional divergence (Figure 1).

### 2.2. Structural and Conserved Motif Analysis of SAUR Proteins

We constructed exon-intron pattern maps of all 116 *PbSAUR* genes to better understand the structural diversity of *SAUR* genes in Dangshansu pear (Figure 2). Among the 116 *PbSAUR* genes, most of them were intronless genes, and only 5 genes (*Pbr003402.1*, *Pbr009914.1*, *Pbr028075.1*, *Pbr036005.1*, *Pbr037398.1*) were intron-containing genes, and *PbrSAUR98* had three introns. These 116 *SAUR* genes have high similarity, which can help us better understand the structural characteristics of *SAUR* genes. We used MEME software to identify 5 conserved motifs, most of the *SAUR* genes contained motif1,2,3 (Appendix A).

### 2.3. Analysis of Cis-Acting Elements in the Promoter of PbrSAUR Gene

The *PbrSAUR* gene is involved in multiple processes of plant growth and development, and a variety of hormones affect its expression. We predicted the *cis*-acting elements for 116 genes of the *PbrSAUR* gene family (Appendix A, Figure 3). The expression of 116 genes in the *PbrSAUR* gene family might be affected by light. These genes contain a variety of light-responsive elements, including 1071 light response elements, such as G-box, I-box, ACE and GT1-motif, of which the number of G-box was the largest. A total of 33 *PbrSAUR* genes contained TC-rich repeats elements, which are involved in defense and stress response. In addition, the *PbrSAUR* gene had many *cis*-acting elements related to hormone response, including gibberellin response element (GARE-motif, P-box, TATC-box), salicylic acid response element (TCA-element, SARE), cis-acting element of abscisic acid response (ABRE) and auxin response element (AuxRR-core, TGA-element). AuxRR-core element related to auxin was found in only one gene (*Pbr009265.1*), which was the least of the elements found. In total, 314 *cis*-acting regulatory elements involved in the MeJA (Methyl Jasmonate) response were found in 79 genes. Low-temperature-responsive *cis*-acting element LTR was found in 36 *PbrSAUR* genes implicated in low-temperature responsiveness. In addition, CAT-box and NON-box were found in 48 *PbSAUR* genes, mainly responding to meristem expression. Among the 116 *PbSAUR* genes, the genes involved in defense and stress response were the least (33 *PbrSAUR* genes); the genes involved in light response were the most, with 116 genes.

### 2.4. Chromosomal Location, Environmental Selection Pressure and Duplication Events of SAUR Family Genes in ‘Dangshansu Pear’

The precise chromosomal position of 116 *SAUR* genes was identified using the full genome sequence of the ‘Dangshansu pear’ and 11 genes could not be located on the chromosome due to the lack of information in the gene annotation file. Most of *PbSAUR* genes (39) were located on chromosome 10, and the least (1) *PbSAUR* genes were located on chromosomes 11 and 16.

The duplication types of the *PbrSAUR* genes were mainly tandem duplication (TD), proximal duplication (PD), dispersed duplication (DSD), transposition duplication (TRD), and whole-genome duplication (WGD). However, the whole genome duplication types were the most abundant and the tandem duplications were the least (Figure 4). This analysis indicates that dispersed duplication plays an important role in the expansion of *SAUR* gene family. To clarify the driving forces of gene duplication and explore the impact of these genes on evolutionary processes, we calculated the Ka, Ks, and Ka/Ks ratios of 21 sets of duplicated gene pairs in ‘Dangshansu pear’. Ka/Ks = 1 means neutral selection, Ka/Ks < 1 means negative selection, and Ka/Ks > 1 means positive selection [23]. Among the 21 gene pairs identified in ‘Dangshansu pear’, most of the gene pairs had a Ka/Ks value were less than 1, and only 3 gene pairs had a Ka/Ks value greater than 1, namely *PbrSAUR9-PbrSAUR8*, *PbrSAUR97-PbrSAUR96*, *PbrSAUR97-PbrSAUR95* (Appendix A). During the evolutionary processes of genes, *PbrSAUR* may experience more negative selection than positive selection [24].

### 2.5. GO Analysis of SAUR Gene Family in ‘Dangshansu Pear’

The online tool eggNOG-mapper can conduct a GO analysis on all members of the *PbrSAUR* gene family to determine the biological activities of each member (Figure 5). Most of the *PbrSAUR* gene family members had no GO annotations, only nine *PbrSAUR* genes have GO annotations, and these nine *PbrSAUR* genes are all involved in the composition of cells, only *PbrSAUR91* (*Pbr007631.1*) had molecular functions and participates in catalytic reactions (Appendix A). The results showed that the *PbrSAUR* gene was mainly involved in the auxin response, and an individual *PbrSAUR* gene (*PbrSAUR6*) provided energy for cell growth and metabolism.

### 2.6. Expression Characteristics of SAUR Gene in ‘Dangshansu Pear’

In order to further understand the ‘Dangshansu pear’ *SAUR* gene family, we investigated the expression patterns of *PbrSAUR* at different developmental stages (15 DAF, 30 DAF, 55 DAF, 85 DAF and 115 DAF) of ‘Dangshansu pear’ fruit (Appendix A). It was found that *PbrSAUR75*, *PbrSAUR65*, *PbrSAUR52*, *PbrSAUR74*, *PbrSAUR13*, *PbrSAUR68* and *PbrSAUR82* had higher expression levels in 55 DAF pear fruits, among which *PbrSAUR13,52* had the highest expression levels. In this study, these two genes were selected for subsequent functional verification.

### 2.7. Analysis of Subcellular Localization

Plant-mPLoc was used to predict subcellular localization of *PbrSAUR13,52* in cells, and it was discovered that they were localized in the nucleus (Appendix A). To verify the accuracy of localization, the localization of *PbrSAUR13*,*52*-GFP recombinant protein was observed by confocal fluorescence microscope with fresh onions (Figure 6). Driven by the 35 S promoter, the expression of the *PbrSAUR13*,*52*-GFP fusion protein was detected in the nucleus, indicating that the protein of *PbrSAUR13*,*52* was mainly localized in the nucleus, which was consistent with the predicted results.

### 2.8. Histochemical Staining Observation of Overexpression of PbrSAUR13,52 in Pear Fruit

We obtained three fresh samples (pear fruits injected with pCAMBIA1304-*PbrSAUR13,52*, and pCAMBIA1304) were first stained with phloroglucinol. We found that there were significant differences in phloroglucinol-stained pear fruits. As shown in Figure 7, the pear fruit injected with pCAMBIA1304-*PbrSAUR52* showed more redness, and the pear fruit injected with pCAMBIA1304-*PbrSAUR13* showed less redness, compared with the empty vector.

Pear fruits with similar growth were selected for safranin fast green staining observation (Figure 8). The staining results were similar to those of phloroglucinol staining. Compared with the control group, the number of stone cell clusters in *PbrSAUR13* fruit was less, and the size of stone cell clusters had little difference. More stone cell clusters were stained red in *PbrSAUR52* fruits, and stone cell clusters were larger than those in the control group.

### 2.9. Analysis of Stone Cells and Lignin Content in Pear Fruit Transiently Transformed with PbrSAUR13,52

The stone cells and lignin contents of transiently transformed pear fruits were measured after chemical staining observation. These results are shown in Table 1 and Figure 9. Compared with the empty vector, the content of stone cells and lignin in pear fruits overexpressing *PbrSAUR13* was lower, the content of the stone cells reached 14.75%, and the lignin content reached 2.99%; The content of stone cells and lignin in pear fruits overexpressing *PbrSAUR52* was higher, the content of stone cells was 19.51%, and the content of lignin was 5.17%.

### 2.10. Expression Pattern Analysis of Key Enzyme Genes in Lignin Biosynthesis in Pear Fruit Transiently Transformed with PbrSAUR 13, 52

The expression patterns of key enzyme genes in lignin synthesis in pear fruits with overexpressing of *PbrSAUR13* and *PbrSAUR52* were analyzed (Figure 10). The results showed that the expression of key enzyme genes of lignin synthesis in pear fruits was inhibited by transient overexpression of *PbrSAUR13*. Among them, the inhibition of genes such as *PbHCT2*, *PbHCT50*, *PbCCR2*, *PbCOMT1* was obvious, and the gene expression level was only 17.6–25% of that in the control group. However, transient overexpression of *PbrSAUR52* activated the expression of most key enzyme genes for lignin synthesis in pear fruits. Among them, *PbHCT49*, *PbPAL3*, and *PbC4H3* were positively regulated, with the highest expression level being increased by 6.74 times. However, the activation effect of the expression levels of *PbHCT50*, *PbPAL1*, *PbCCR2*, *PbCCoAOMT*, and *PbCOMT1* was not obvious, which was only 90% of the control group.

### 2.11. Infrared Spectrum Analysis of Lignin in ‘Dangshansu Pear’

It can be seen from Figure 11 and Table 2 and Table 3, the infrared spectrum at 1251 cm^−1^ is the characteristic peak of the guaiac ring plus C=O stretching vibration, the characteristic peak at 1328 cm^−1^ was the characteristic peak of syringyl, and the characteristic peak at 1251 cm^−1^. The characteristic peak higher than 1328 cm^−1^ indicated that the characteristic peak of guaiac ring plus C=O stretching vibration was higher than that of syringyl. The change in the relative content of G-lignin and S-lignin (G/S) could be expressed as the change in the ratio of A1251/A1328. The results revealed that the content of G-lignin in ‘Dangshansu pear’ was higher than that of S-lignin. Compared with pCAMBIA1304 empty load, the generation intensity of G-lignin and S-lignin in pear fruit transiently transformed with *PbrSAUR13* decreased to 1.476 and 1.0825, respectively, and G/S also decreased to 1.3286 (Table 3). The generation intensities of G-lignin, S-lignin and G/S in pear fruit transiently transformed with *PbrSAUR52* all increased, reaching 1.6188, 1.1375 and 1.4232, respectively.

### 2.12. Analysis of Lignin Content in Strawberry Transiently Transformed with PbrSAUR13 and PbrSAUR52

Strawberry belongs to the Rosaceae species, and the pulp is white. If the lignin content changes, the observation will be more obvious, and strawberry is often used as a model crop of woody plants to determine the change in lignin content. To further verify the function of *PbrSAUR13* and *PbrSAUR52*, we then measured the lignin content of overexpressed strawberry fruit (Figure 12). Similarly, we also selected three samples (strawberries injected with pCAMBIA1304-*PbrSAUR13,52*, and pCAMBIA1304). The lignin content of strawberries injected with pCAMBIA1304 empty was about 0.384%. In strawberry overexpressed by *PbrSAUR13*, the lignin content is about 0.381%, compared with the control group, the lignin content in strawberries overexpressing by *PbrSAUR52* will be increased by about 0.398%.

### 2.13. Expression Pattern Analysis of Key Enzyme Genes in Lignin Biosynthesis in Strawberry Transiently Transformed with PbrSAUR 13, 52

In strawberries overexpressed with *PbrSAUR13* and *PbrSAUR52*, the expression patterns of important enzyme genes for lignin production were studied (Figure 13). The results showed that transient overexpression of *PbrSAUR13* inhibited the expression of key enzyme genes of lignin synthesis in strawberries, among which the inhibition of *POD*. The effect was obvious, and the gene expression level was only 1.75% of that in the control group. However, transient overexpression of *PbrSAUR52* could activate the expression of key enzyme genes of lignin synthesis in strawberry, especially *POD*, and the gene expression level was increased by 10.74 times compared with the control group.

## 3. Discussion

‘Dangshansu pear’ is produced in Dangshan County, Anhui Province. It is the pear variety with the largest cultivation area in China. The inherent defects of high content and large diameter of stone cells are increasingly prominent, which seriously affects its quality and processing. To improve the quality of pear fruit, researchers began to study the accumulation and biosynthesis of lignin in pear [18,25,26]. The auxin early response factor (*SAUR*) is a class of genes most rapidly and violently responsive to auxin, which promotes cambium-specific activity and leads to the development of lignin-rich conducting tissue. However, the effect of *SAUR* gene on stone cells and lignin was still unclear.

In this study, we identified 116 *SAUR* genes from ‘Dangshansu pear’ (Appendix A). Among 63 genes, the isoelectric points (pIs) ranged from 5.1 to 10.28 with an average of 7.77 and their GRAVY values are less than 1, *PbrSAUR* proteins are all hydrophilic proteins [27]. Gene structure and the composition of conserved sequences might influence gene function [28]. Genes in the same branch of the evolutionary tree tend to have similar gene structures and conserved sequences, suggesting that these genes may have similar functions. For example, *PbrSAUR47* (*Pbr009936.1*) and *PbrSAUR79* (*Pbr042223.1*) have similar gene structures, both only have exons and are intronless genes. Additionally, the conserved sequences of these two genes are also highly uniform, and both have four motif elements (Motif 1, 2, 3, and 4), and the arrangement of motif elements is highly similar. Genes can lose their original functions and introduce new functionalities as a result of gene duplication events [29]. In addition, gene duplication events are the main driving force for gene family expansion, which is conducive to plants’ adaptation to environmental changes [30]. There were 21 pairs of replication gene pairs in ‘Dangshansu pear’, which promoted the expansion of the *SAUR* gene family and enriched the functions of the *PbrSAUR* gene family. We performed Ka and Ks analysis on 21 replicated gene pairs of ‘Dangshansu pear’, and the results showed that most gene pairs were negatively selected, and only 3 gene pairs had undergone purification selection. The *PbrSAU*R gene family is mainly selected by purification to maintain the stability of gene function. In current study, the *PbrSAUR13* and *PbrSAUR52* genes were screened out by bioinformatics analysis of the *PbrSAUR* gene family. The 39 DAF ‘Dangshansu pear’ fruit and the ginkgo stage Frandi strawberry were selected for further expression analysis. The pCAMBIA1304-*PbrSAUR13*,*52* and pCAMBIA1304 empty agricultural Bacillus suspension was injected into fruit material, and the content of lignin in stone cells was determined. It was found that *PbrSAUR13* significantly inhibited the synthesis of lignin in ‘Dangshansu pear’ fruit and strawberry, and *PbrSAUR52* promoted the synthesis of lignin in pear fruit and strawberry.

Stone cells and lignin in the fruit of ‘Dangshansu pear’ were mainly synthesized during 15–63 DAF [31]. Referring to the transcriptome data of ‘Dangshansu pear’ fruit before and after ripening, it was found that *PbrSAUR13* and *PbrSAUR52* were differentially expressed before and after pear fruit ripening. It was speculated that these two genes affected the accumulation of lignin in pear fruit stone cells. The subcellular localization of these two genes showed that both genes were localized in the nucleus, which was consistent with the predicted results. *PbrSAUR13* and *PbrSAUR52* were transiently transformed into pear fruit, and the pear fruits were stained with phloroglucinol, and the content of stone cell lignin was determined. The results exhibited that compared with the control group, the contents of stone cells and lignin in pear fruits transiently transformed with *PbrSAUR13* were significantly lower. In contrast, the contents of stone cells and lignin in pear fruits transiently transformed with *PbrSAUR52* were higher than those in the control group. Therefore, we speculate that *PbrSAUR13* may inhibit the synthesis of stone cells and lignin, and *PbrSAUR52* may promote the synthesis of stone cells and lignin in pear fruit. According to the composition of lignin polymers, lignin can be divided into guaiacyl lignin (G-type lignin), p-hydroxyphenyl lignin (H-type lignin), lilac Syringyl lignin (S-type lignin) [21]. In order to study the effect of *PbrSAUR13,52* gene overexpression on pear fruit lignin polymer monomers, we used Fourier transform infrared spectroscopy to analyze the lignin structure by infrared spectroscopy. These results showed that compared with the control group, the contents of G-lignin and S-lignin in pear fruit transiently transformed by *PbrSAUR13* decreased, and G/S also decreased; while *PbrSAUR52* transiently transformed pear fruit G-lignin, S-lignin and G/S all increased.

In the current study, *PbrSAUR13,52* was transiently transformed into Frandi Strawberry to study the effects of these two genes on the lignin of strawberry fruit. The results showed that in strawberries transiently transformed with *PbrSAUR13*, the expression of key enzymes in strawberry lignin synthesis was inhibited, and compared with the control group, the lignin content was reduced; in strawberries transiently transformed with *PbrSAUR52*, the expression of key enzymes in strawberry lignin synthesis was activated, and the lignin content significantly increased.

## 4. Materials and Methods

### 4.1. Identification and Characterization of SAUR Gene in Pyrus bretschneideri

In this study, the CDs sequence, protein sequence, genome sequence, and gene annotation file (GFF/gff3 format) of *Pyrus bretschneideri* were obtained from the Rosaceae species database GDR (https://www.rosaceae.org/, accessed on 1 June 2021) [32]. DNATOOLS software was used to create a local database of all gene amino acid sequences of *Pyrus bretschneideri* [33]. Referring to Molinari [34], the *SAUR* family characteristic domain (Pfam: PF02519) was used as the query sequence, and DNATOOLS [35] software was used for comparison and screening in *Pyrus bretschneideri* genome database (E-value = 0.001), and the candidate *SAUR* sequences were preliminarily screened. The candidate sequences were analyzed online by SMART [36] (http://smart.embl-heidelberg.de/, accessed on 12 June 2021) and Pfam [37] (http://pfam.xfam.org/, accessed on 20 June 2021) software to identify whether they had *SAUR* family characteristics domains. Finally, the sequence of *SAUR* family members was obtained. The molecular weight of *SAUR* protein was predicted by ExPASY online website (http://web.expasy.org/protparam/, accessed on 1 July 2021) [38]. The subcellular localization of all *SAUR* proteins was predicted by the Wolf PSORT website (http://www.genscript.com/wolf-psort.html, accessed on 20 July 2021) [39].

### 4.2. Phylogenetic Analysis

Sequence alignment of all SAUR proteins were sequenced using the ClustalW tool in MEGA-X software. The phylogenetic tree was constructed with MEGA-X software using the maximum likelihood (ML) (bootstrap = 1000) [40]. The SAUR gene sequences of Arabidopsis thaliana and Oryza sativa used as the reference genome in the phylogenetic tree were derived from the results of Wang et al. [41]. Finally, the phylogeny was visualized using itols software (https://itol.embl.de/login.cgi, accessed on 5 August 2021).

### 4.3. SAUR Gene Structure and Conserved Motif Prediction

Comparison of indeterminate domain gene structure was carried out through Gene Structure Display Server (http://gsds.cbi.pku.edu.cn, accessed on 15 August 2021) [42]. The motifs of the *SAUR* genes in *Pyrus bretschneideri* were analyzed by MEME online analysis software (http://meme.sdsc.edu/meme4_3_0/intro.html, accessed on 27 August 2021) [43]. The parameters of conservative motifs prediction were greater than 6 and less than 200 of motif width and set the number of recognized motifs was 1 to 5.

### 4.4. Chromosome Mapping, Gene Duplication Events, and Ka/Ks Ratio Analysis

The chromosome starting position and the related position information of *SAUR* gene were obtained from the genome annotation file of *Pyrus bretschneideri*. The chromosome physical locations of the *SAUR* genes were mapped by MapInspect software (http://mapinspect.software.informer.com, accessed on 9 September 2021) [44,45,46]. Genes and duplicates were ascribed to one of five different duplication categories: transposed, tandem, proximal, WGD, and dispersed [47]. Non-synonymous (Ka) and synonymous substitution (Ks) values were calculated using TBtools software [48].

### 4.5. Analysis of Cis-Acting Elements of SAUR Gene Promoter in Pyrus bretschneideri

We obtained the promoter sequence of each *SAUR* gene from the genome sequence of *Pyrus bretschneideri*, including the DNA sequence of the start codon (ATG) located 2000 bp upstream of each *SAUR* gene. The online software PlantCARE was used to analyze the *cis*-acting elements of the promoter region (http://bioinformatics.psb.ugent.be/webtools/plantcarere/html/, accessed on 22 September 2021) [49].

### 4.6. GO Functional Enrichment Analysis of SAUR Gene Family in Pyrus bretschneideri

The protein sequence of SAUR gene family of Pyrus bretschneideri was annotated by eggNOG-mapper tool, and the analysis results were analyzed by WEGO—a web tool for plotting GO annotations [50] (https://wego.genomics.cn/, accessed on 2 October 2021) for statistics and visualization.

### 4.7. Gene Screening and Transcriptomic Analysis

*P. bretschneideri* fruit of 15 DAF, Accession: SRX1595645; *P. bretschneideri* fruit of 30 DAF, Accession: SRX1595646; *P. bretschneideri* fruit of 55 DAF, Accession: SRX1595647; *P. bretschneideri* fruit of 85 DAF, Accession: SRX1595648; *P. bretschneideri* fruit of 115 DAF, Accession: SRX1595650; *P. bretschneideri* fruit at the mature stage, Accession: SRX1595651; *P. bretschneideri* fruit at senescence stage, Accession: SRX1595652 were downloaded from NCBI website (https://www.ncbi.nlm.nih.gov/, accessed on 3 January 2022). TBtools was used for heat map screening of subsequent genes for functional verification.

### 4.8. Gene Cloning and Construction of Plant Expression Vectors

According to the full-length sequence of *PbrSAUR13,52*, the full-length sequence-specific primers were designed by Primer Premier 6.0 software. The cDNA of *P. bretschneideri* fruit was used as a template for RT-PCR. Primers with restriction sites using Primer Premier 6.0 software (pCAMBIA1304-*PbrSAUR13*) used Bgl II and Spe I restriction sites, pCAMBIA1305-*PbrSAUR13* used XbaI and SmaI restriction sites, pCAMBIA1304-*PbrSAUR52* used XbaI and BamHI restriction sites, and pCAMBIA1305-*PbrSAUR52* used Xba I and Sma I restriction sites) (Appendix A). Each gene fragment was ligated with pCAMBIA1304 and pCAMBIA1305 vectors by T4 ligase at 16 °C for 3 h to obtain complete pCAMBIA1304-*PbrSAUR13,52* and pCAMBIA1305-*PbrSAUR13,52* recombinant plasmids. 

### 4.9. Plant Materials

The materials used in this study are ‘Dangshan Su pear’ grown in the former Yichang Agricultural Park in Dangshan County, Anhui Province, China, and ‘Flandi’ strawberry from Yanjiutian Strawberry Base in Changfeng County, Hefei City, Anhui Province. According to the previous research in the laboratory, we found that in the 39 DAFs (day after flowering) pear fruit and the ginkgo stage strawberry, the lignin content changed significantly, so we chose ‘Dangshan Su pear’ of 39 DAF and ‘Flandi’ strawberry to ginkgo stage were selected, and pCAMBIA1304-*PbrSAUR13* and *PbrSAUR13,52* and pCAMBIA1304 empty Agrobacterium suspension were prepared to inject Dangshan pear and Frandi strawberry. The injected materials were picked back after 7 days. Some fresh pear fruits are selected for dyeing observation to determine the content of stone cells and lignin. The rest of the materials were stored in the −80 °C refrigerator. Some fresh strawberries were selected to determine the lignin content, and their materials were stored in the −80 °C refrigerator for further analysis.

### 4.10. Extraction of Total RNA from Plant Material and qRT-PCR

The plant RNA extraction kit V1.5 of Chengdu Best Technology Co., Ltd. was used to extract the 39 DAF fruit RNA of Dangshansu pear, and the plant RNA extraction kit V1.6 of Chengdu Best Technology Co., Ltd. located in Chengdu, China was used to extract ‘Frangi’ strawberry fruit. RNA was reverse transcribed into cDNA using the Easy Script One-Step gDNA Removal and cDNA Synthesis Super Mix kit from Beijing Quanshijin Biotechnology Co., Ltd. located in Beijing, China. All qRT-PCR primers in this study were designed using Primer Premier 5 software, and liquid primers were ordered from Sangon Bioengineering (Shanghai, China) (Appendix A). The qRT-PCR system consists of 10 μL SYBR^®^ Premix Ex TaqTM II (2X), 6.4 μL water, 0.8 μL upstream and downstream primers, and 2 μL cDNA (20 μL total). For the qRT-PCR experiments, we chose the tubulin gene (accession AB239680.1) as the internal reference gene [51]. During qRT-PCR, each gene was assessed by three biological replicates, and the 2^−ΔΔCT^ method was used to process the data and calculate the relative expression levels of the genes [52].

### 4.11. Subcellular Localization Analysis

Subcellular localization prediction of genes was performed by Plant-mPLoc, and then the Agrobacterium transferred into pCAMBIA1305-*PbrSAUR13,52* recombinant plasmid was enriched and cultured according to the method of Xu et al. [53]. The roots of Agrobacterium suspension, a base of ½ MS, MES (10 mM), Silwett L-77 (0.01%), MgCl_2_ (0.5 mM), AS (200 μM), roots of agrobacterium suspension, the OD of agrobacterium suspension 600 = 0.2, The cells were incubated at room temperature (50 rpm) for 1 h. Select fresh onions, chop them gently, remove the first two layers of scales, take about 1 cm^2^ of the lower epidermis from the middle scale (the third layer of scales or internal scales), immerse in the agricultural suspension medium, and culture in the dark for 30 min. According to ½ MS Sucrose (1%), Casein (0.03%), Proline (0.28%), 2, 4-D (10 μM), BAP (2 μM), AS (200 μM), Agar (0.8%), pH 5.8 formula configuration co-culture plate. After the infection of the onion lower epidermal cells, the epidermis was dried with sterilized filter paper (slightly removing the excess agricultural suspension medium), then transferred to a separate co-culture medium plate and cultured in the dark at 22 °C for 20–24 h.

### 4.12. Observation of Phloroglucinol Staining in ‘Dangshansu Pear’ Fruit

The samples of pear fruit were collected from healthy and disease-free plants. After slitting the fruit, put the longitudinal section into 10% phloroglucinol solution for 30 s, wipe off the excess phloroglucinol solution, and put it into 1 mol/L hydrochloric acid solution, observe the dyeing condition after the 30 s and record with a camera.

### 4.13. Observation of the Staining of Fruit Tissue Sections of ‘Dangshansu Pear’

Pick pear fruit samples that are healthy and uniform size clean them, put them in FAA fixative solution, and fix them for 12 h. After fixation, the fruits were embedded in wax blocks, sliced on an automatic microtome, and then stained with safranin O-fast green observation with a microscope.

### 4.14. Determination of Stone Cells and Lignin in Pear Fruit

Each sample group was weighed at 5.0 g and frozen at −20 °C for 1 day. The samples were then homogenized at 20,000 rpm for 3 min. Water was added to the homogenate and allowed to stand. After everything sank to the bottom of the beaker, the suspended matter in the upper layer was poured out and repeated several times until the upper solution is clear. Finally, the stone cells were dried and weighed, and the stone cell content of each fruit sample was calculated as follows: stone cell formula weight (g DW)/pulp weight (g FW) × 100 = stone cell content (%) [31].

After removing the skin and core of the pear fruit, it was dried in an oven at 37 °C, and then the dried fruit material was ground into powder and passed through a 20-mesh sieve. The powder was extracted with methanol, and the extracted residue was dried. Next, 0.2 g of the dried residue was weighed and extracted in 15 mL of 70% H_2_SO_4_ for 1 h at 30 °C. A total of 115 mL of distilled water was added, and the solution was boiled for 1 h. During the boiling process, the total volume of the sample remained unchanged. The boiled mixture was filtered with filter paper and rinsed with distilled water at 70 °C until the rinse solution was neutral and clear. The lignin residue was dried and weighed, and each sample was repeated three times.

### 4.15. Lignin Extraction and Purification

The dried lignin was ground into powder under mild conditions, passed through a 200-mesh sieve, and extracted with benzene-ethanol (volume 1:1) at 40 °C. The ratio of powder to benzene-ethanol was (W:V) 1:10.

### 4.16. Determination of Lignin by Infrared Spectroscopy

Referring to the methods of Chang et al. [54]: Weigh 10 mg of purified lignin and mix with 1 g of KBr, grind them with an agate mortar in a dry environment, press them with a tablet machine, and determine them with Fourier transform Measured by an infrared spectrometer, A1251/A1328 can represent the relative content of G-lignin and S-lignin.

### 4.17. Determination of Lignin in Strawberry

Weigh 1 g strawberry fruit powder, add 3 mL of 95% (1:3, *v*/*v*) ethanol solution, and centrifuge at 4 °C for 10 min. The obtained precipitate was washed three times with 95% ethanol, then washed three times with a solution ratio of ethanol:n-hexane = 1:2, then 1 mL of 2 mol/L NaOH was added to stop the reaction, followed by 2 mL of CH3COOH and 1 mL 7.5 mol/L of hydroxylamine hydrochloride, centrifuged for 15 min. Take 0.5 mL of the supernatant, dilute to 4 mL with glacial acetic acid, and determine the absorbance value at 280 nm. The calculation formula is: lignin% = (Abs × liters × 100%)/(W sample × A standard). Where “lignin%”: lignin content; “Abs”: the absorbance of the sample solution at 280 nm; “liters”: the volume of the sample solution at constant volume (L); “Wsample”: the absolute dry weight of the sample (g); “Astandard”: Arabidopsis lignin standard absorbance 17.2. The unit is expressed as OD280 g^−1^. Each sample was repeated three times.

## 5. Conclusions

Total 116 *SAUR* genes were identified in ‘Dangshansu pear’ using bioinformatics analysis and these genes were classified into four groups based on the phylogeny. Within each taxonomy, genes were very similar in structure and conserved sequences. According to the analysis of promoter *cis*-acting elements, it was found that all *PbrSAUR* gene families were involved in light response, and the genes involved in defense and stress response were the least, with only 33 *PbrSAUR* genes. Through the transient transformation of 39 DAF ‘Dangshansu’ pear fruit and ginkgo-stage Frandi strawberry and found that both *PbrSAUR13,* and *PbrSAUR52* were involved in lignin regulation. *PbrSAUR13* inhibited the synthesis and accumulation of stone cells and lignin, while *PbrSAUR52* mainly promoted the synthesis and accumulation of stone cells and lignin.

## Figures and Tables

**Figure 1 ijms-23-07054-f001:**
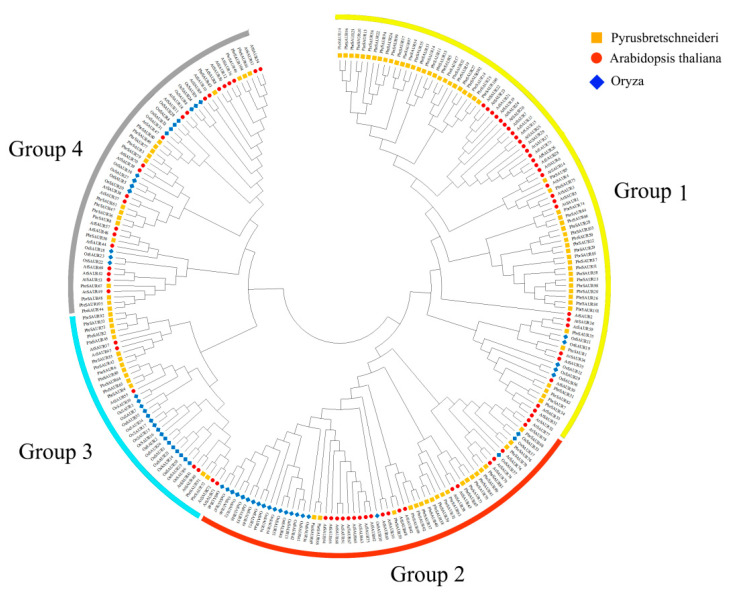
Phylogenetic relationships and subfamilies in SAUR proteins of ‘Dangshansu pear’ (*Pyrus bretschneideri*), *Arabidopsis thaliana,* and *Oryza sativa*. Group 1–4 are represented by yellow, red, blue, and gray, respectively.

**Figure 2 ijms-23-07054-f002:**
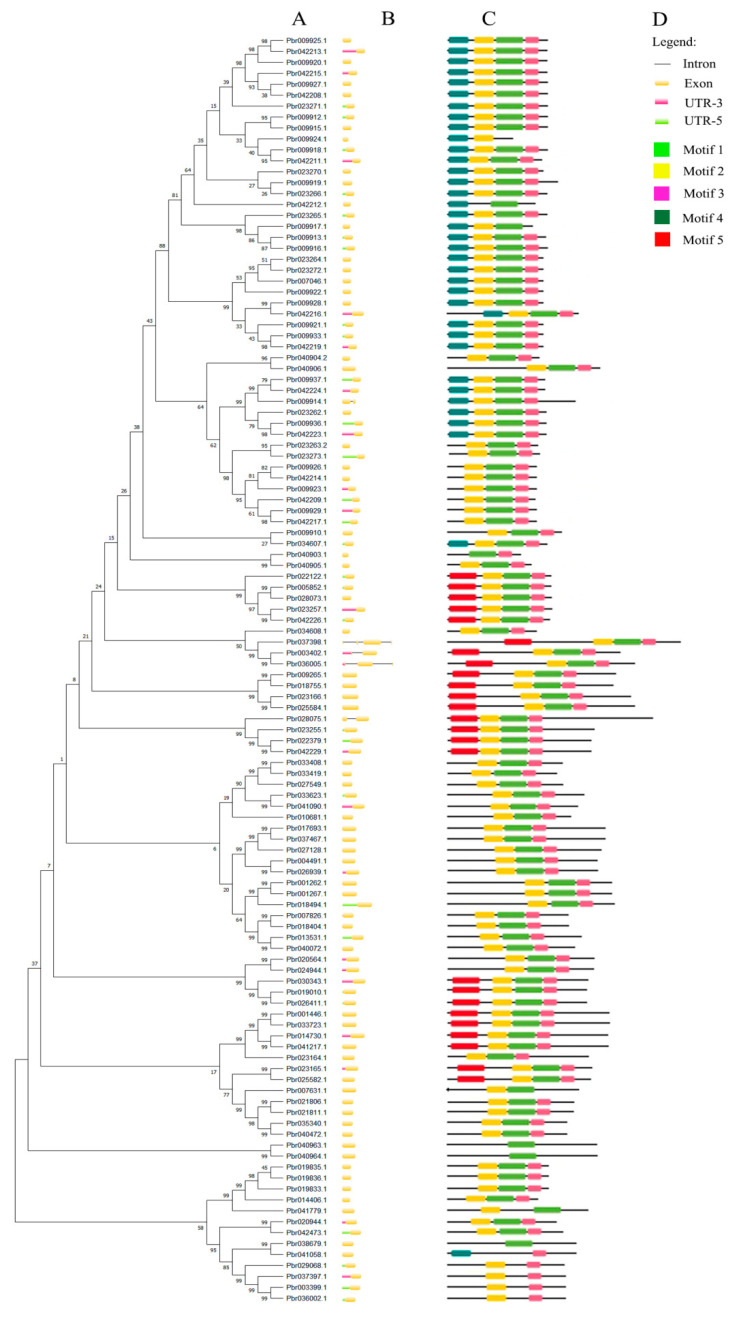
Predicted the conserved motif and exon-intron structure of ‘Dangshansu pear’ *SAUR* gene. (**A**) Phylogenetic tree of *PbrSAUR* family members; (**B**) exon-intron structure of *PbrSAUR* family members, black lines indicate introns and yellow wedges indicate exons; (**C**) conserved elements of *PbrSAUR* family members; (**D**) the color and a corresponding number of each motif box.

**Figure 3 ijms-23-07054-f003:**
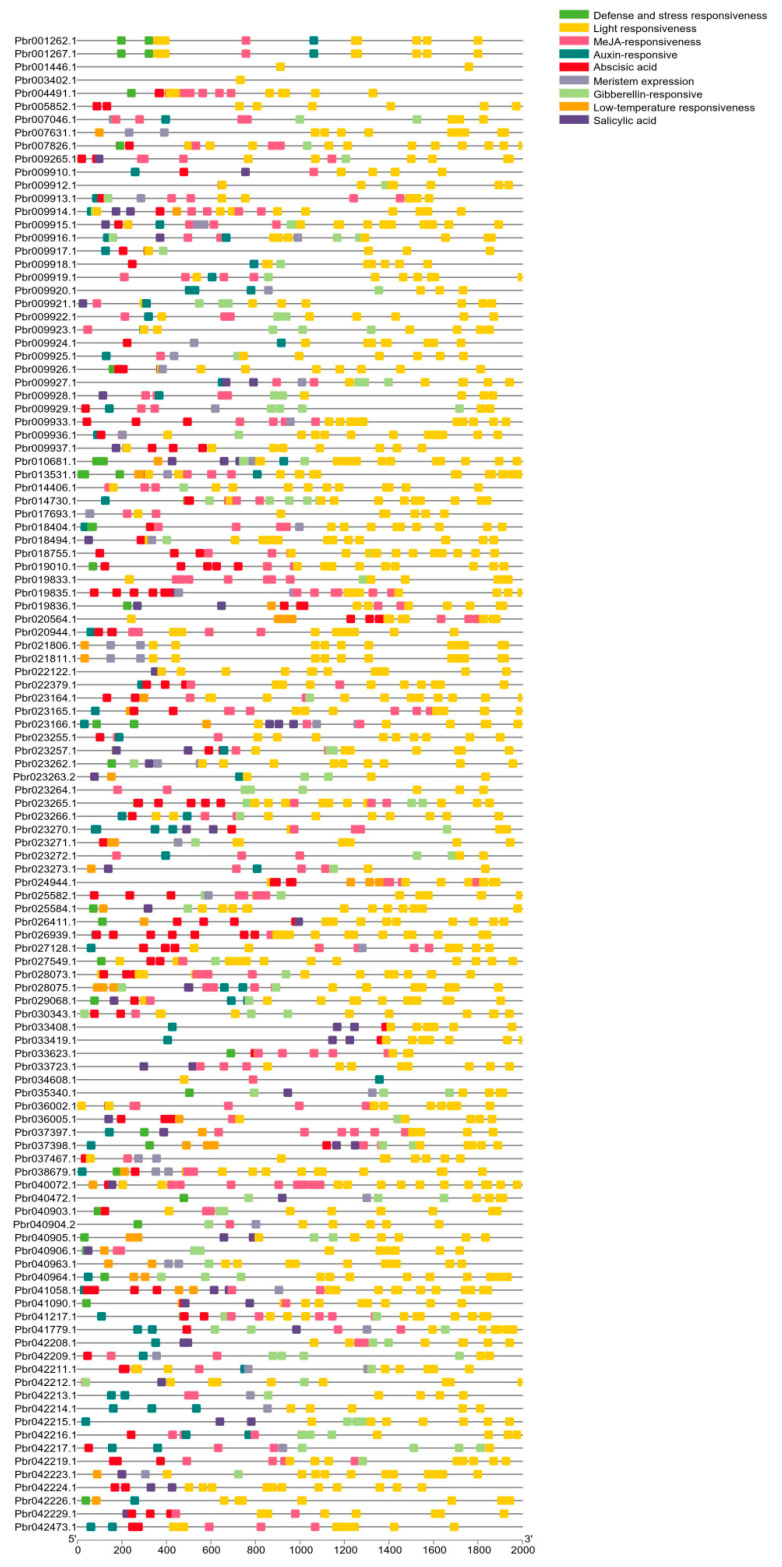
Promoter *Cis*-elements of the 116 *PbrSAURs*.

**Figure 4 ijms-23-07054-f004:**
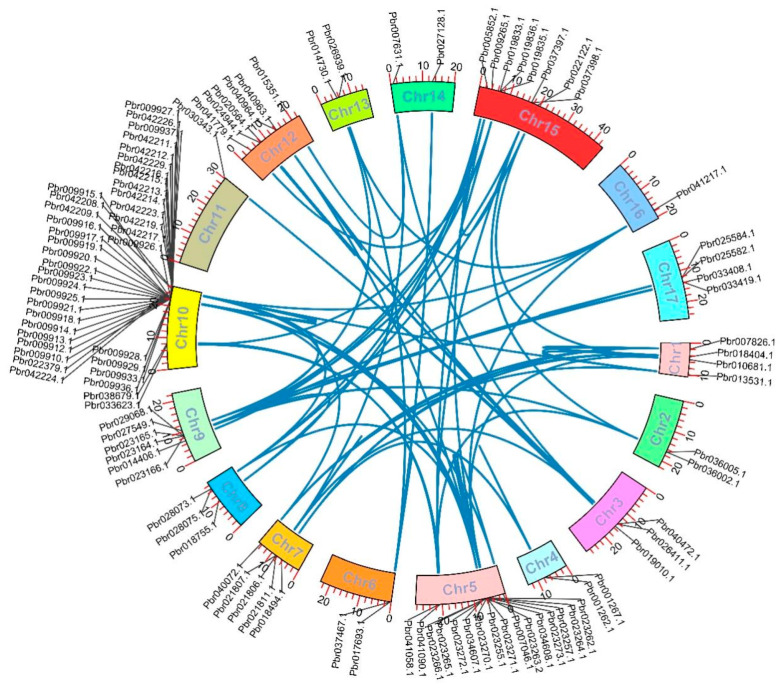
Analysis of duplication type and chromosomal localization of *SAUR* gene family in ‘Dangshansu pear’. Duplicated gene pairs are linked with a blue line.

**Figure 5 ijms-23-07054-f005:**
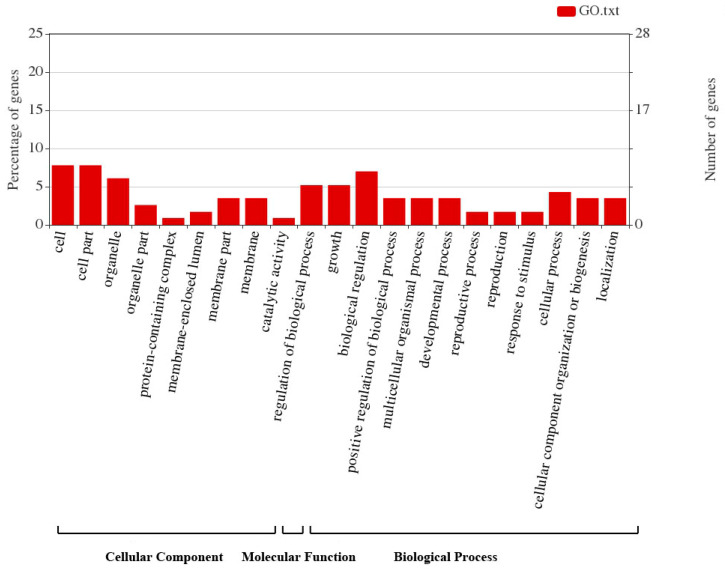
GO annotation analysis of *PbrSAUR* gene family and these functions divided into three groups cellular components, molecular function, and process.

**Figure 6 ijms-23-07054-f006:**
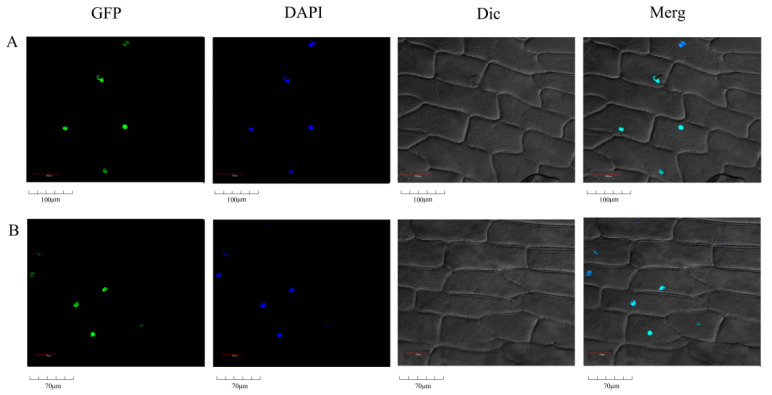
Subcellular localization of *PbrSAUR13*-GFP and PbrSAUR52-GFP fusion protein. (**A**) Subcellular localization of *PbrSAUR13*-GFP fusion protein; (**B**) Subcellular localization of P*brSAUR52*-GFP fusion protein.

**Figure 7 ijms-23-07054-f007:**
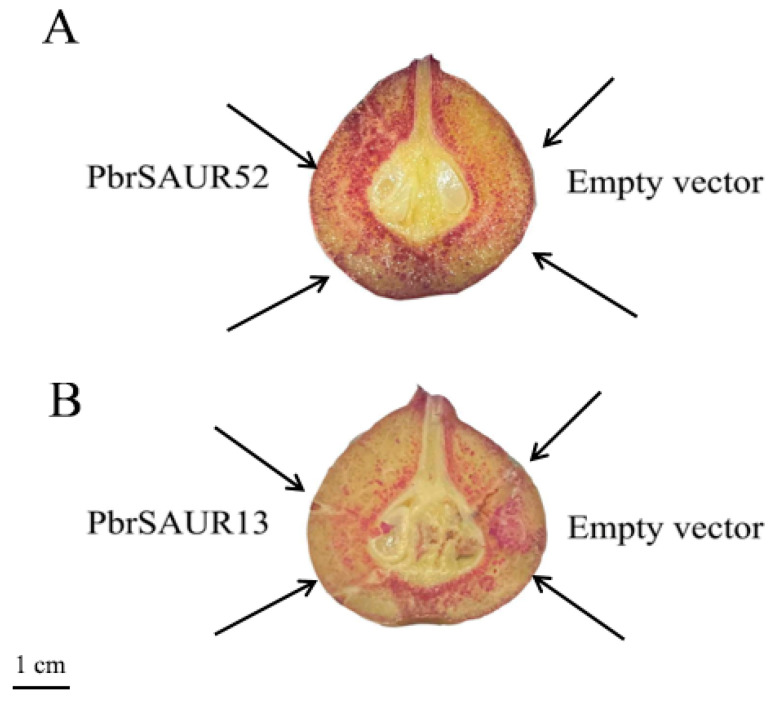
Phloroglucinol staining of ‘Dangshansu pear’ fruit overexpressed by *PbrSAUR13* and *PbrSAUR52*, empty vector: pCAMBIA1304. (**A**) Phloroglucinol staining in cross-sections of pear fruit transiently overexpressing *PbrSAUR52*; (**B**) phloroglucinol staining in cross-sections of pear fruit transiently overexpressing *PbrSAUR13*.

**Figure 8 ijms-23-07054-f008:**
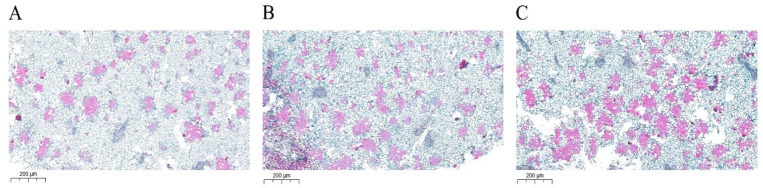
Observation of safranin-fast green staining in fruits of “Dangshansu” pear transiently overexpressed by *PbrSAUR13,52.* (**A**) Safranin fast green staining of pCAMBIA1304 empty pear fruit; (**B**) Safranin fast green staining in pear fruit overexpressing by *PbrSAUR13*; (**C**) Safranin fast green staining in pear fruit overexpressing by *PbrSAUR52*.

**Figure 9 ijms-23-07054-f009:**
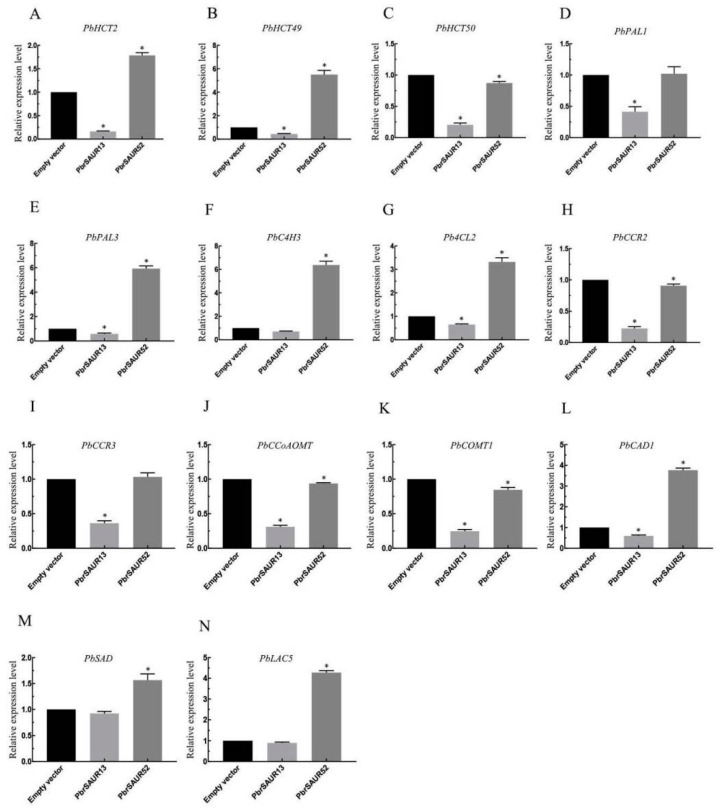
Analysis of gene expression patterns of key enzymes in lignin synthesis in pear fruits overexpressing *PbrSAUR13*,*52*. * Significant difference at *p* < 0.05. 14 key enzyme genes (**A**) *PbHCT2*, (**B**) *PbHCT49*, (**C**) *PbHCT50*, (**D**) *PbPAL1*, (**E**) *PbPAL3*, (**F**) *PbC4H3*, (**G**) *Pb4CL2*, (**H**) *PbCCR2,* (**I**) *PbCCR3*, (**J**) *PbCCoAOMT*, (**K**) *PbCOMT1*, (**L**) *PbCAD*1, (**M**) *PbSAD*, (**N**) *PbLAC5* of lignin synthesis in pear fruits.

**Figure 10 ijms-23-07054-f010:**
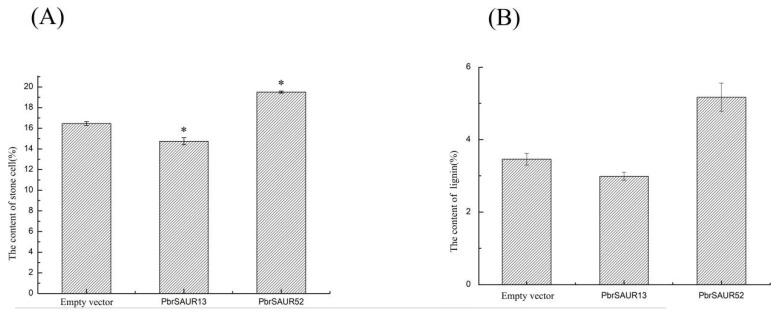
The contents of stone cells (**A**) and lignin (**B**) of *PbrSAUR13,52* overexpressed pear fruits. * Significant difference at *p* < 0.05.

**Figure 11 ijms-23-07054-f011:**
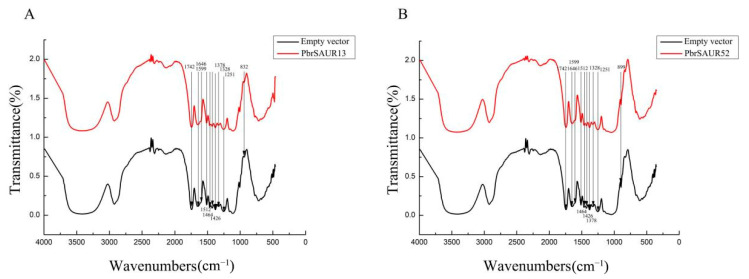
Infrared spectrum of lignin in Pear Fruit overexpressed by *PbrSAUR13* (**A**) and *PbrSAUR52* (**B**).

**Figure 12 ijms-23-07054-f012:**
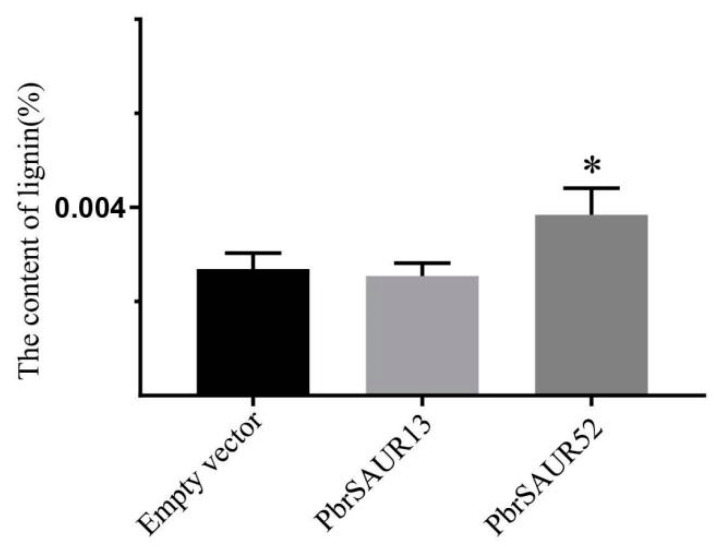
The contents of lignin of *PbrSAUR13,52* overexpressed strawberry. * Significant difference at *p* < 0.05.

**Figure 13 ijms-23-07054-f013:**
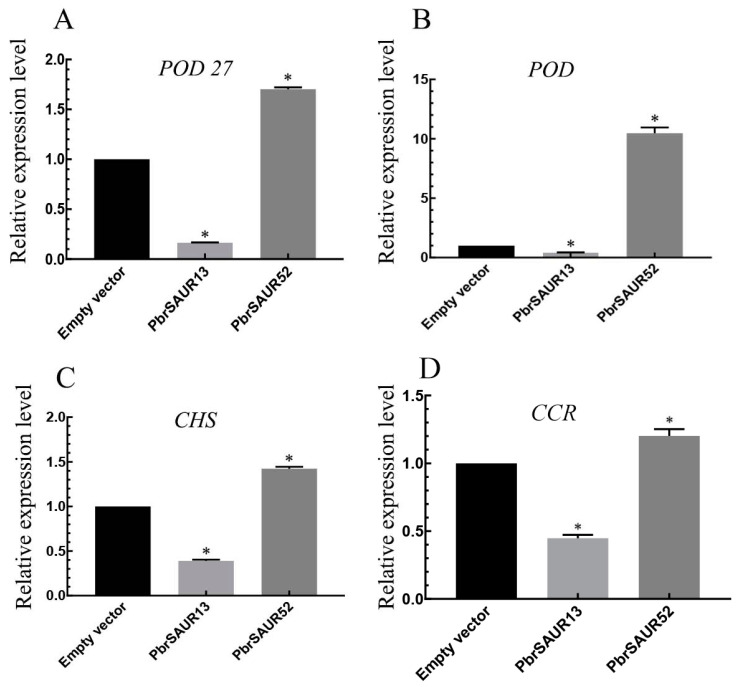
The expression pattern of *PbrSAUR13,52* overexpressing in strawberry. * Significant difference at *p* < 0.05. Key enzyme genes (**A**) *POD27*, (**B**) *POD*, (**C**) *CHS*, (**D**) *CCR* of strawberry lignin synthesis.

**Table 1 ijms-23-07054-t001:** The contents of stone cells and lignin in pear fruit overexpressed by *PbrSAUR13* and *PbrSAUR52*.

	Empty Vector	*PbrSAUR13*	*PbrSAUR52*
Stone cell content (%)	16.46 ± 0.0019	14.75 ± 0.0034 **	19.51 ± 0.0011 *
Lignin content (%)	3.46 ± 0.00165	2.99 ± 0.00112 *	5.17 ± 0.0039 *

NOTE: * Significant difference at *p* < 0.05, ** Significant difference at *p* < 0.01.

**Table 2 ijms-23-07054-t002:** Infrared spectrum characteristic peaks and attribution of lignin in *PbrSAUR13,52* overexpressed pear fruit.

Absorption Peak Range/cm^−1^	Ascription
1742	C=O stretching vibrations in unconjugated ketones, carbonyls and esters
1646	Conjugated carbonyl C=O stretching vibration
1599	Aromatic nuclear stretching vibration plus C=O stretching vibration
1512	Aromatic nucleus vibration
1464	C-H deformation vibration of methyl or methylene
1426	C-H Plane Deformation Vibrations on Aromatic Rings
1378	Aliphatic C-H and phenolic O-H stretching vibrations in CH3 (not Ome)
1328	Syringe base
1251	Guaiac based ring plus C=O stretching vibration
832	Guaiacol-type aromatic nuclei at positions 2.5 and 6. C-H vibrations out of the upper plane

**Table 3 ijms-23-07054-t003:** *PbrSAUR13, PbrSAUR52* overexpressed pear fruit G-type lignin and S-type lignin band intensity ratio.

	Empty Vector	*PbrSAUR13*	*PbrSAUR52*
A1251/A1512	1.5347 ± 0.03493	1.4726 ± 0.02163 *	1.6188 ± 0.01304 *
A1328/A1512	1.0975 ± 0.04292	1.0825 ± 0.01514 **	1.1375 ± 0.00903
A1251/A1328	1.3974 ± 0.05443	1.3286 ± 0.00422 *	1.4232 ± 0.00637 **

NOTE: * Significant difference at *p* < 0.05, ** Significant difference at *p* < 0.01.

## Data Availability

The CDs sequence, protein sequence, genome sequence, and gene annotation file (GFF/gff3 format) of Pyrus bretschneideri were obtained from the Rosaceae species database GDR (https://www.rosaceae.org/, accessed on 1 June 2021). The *SAUR* gene sequences of *Arabidopsis thaliana* and *Oryza sativa* were derived from the results of Phytozome database (https://phytozome-next.jgi.doe.gov/, accessed on 5 June 2021) Transcriptomic data of *P. brestschneideri* was downloaded from the NCBI website (https://www.ncbi.nlm.nih.gov/sra, accessed on 3 January 2022) of different fruit developmental stages with accession number SRX1595645, SRX1595646, SRX1595647, SRX1595648, SRX1595650, SRX1595651, SRX1595652.

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
