# Peer review of "Comparative Genomic Analysis of SAUR Gene Family, Cloning and Functional Characterization of Two Genes (PbrSAUR13 and PbrSAUR52) in Pyrus bretschneideri"

_ijms, 2022, doi:10.3390/ijms23137054_

Round 1

Reviewer 1 Report

This paper reports a comparative genomic analysis of the SAUR gene family among 'Dangshansu pear' (Pyrus 92 bretschneideri), Arabidopsis thaliana and Oryza sativa, and the role of PbrSAUR13 and PbrSAUR52 in lignin synthesis. This work will be helpful for the researcher in understanding the molecular mechanism of lignin biosynthesis and the content of stone cells of pears. However, I have several points that might need the author's attention.

  1. The first paragraph in the result section described the isoelectric points of SAUR proteins. However, I did not see the relevance of the isoelectric to this study as the author did not discuss it in either discussion or introduction section.
  2. In the result section 2.1, SAUR genes number were reported least and most in groups 3 and 1 of the phylogenetic tree, respectively. Is there any evolutionary explanation for this? Please explain in the result or discussion section.
  3. Different Cis-acting elements were identified among SAUR genes, but no validation or confirmation has been provided about the roles of these elements and their impact on the gene expression. Does the author predict any connection between Auxin, PbrSAUR13, and 52, lignin biosynthesis genes?
  4. Please explain whether PbrSAUR13 and PbrSAUR52 impact the number of stone cells or lignin contents only in the stone cells; it is unclear to me.
  5. What was the data source to investigate the expression level of SAUR genes at different developmental stages? Please explain in the result/material and method section.
  6. Each sample was replicated three times, but the Material and Method section did not mention how many samples were used.
  7. Please elaborate on the " excellent growth conditions" mentioned in the manuscript in multiple places. 

Author Response

Title: Comparative Genomic Analysis of SAUR Gene Family, Cloning and Functional Characterization of two genes (PbrSAUR13 and PbrSAUR52) in Pyrus bretschneideri

Dear Editor and Reviewers,

We thank all the Reviewers and the Editor for their careful reading of our manuscript and overall supportive comments and suggestions. This constructive input has contributed to improve the manuscript significantly, where we have also had a special eye to enhance clarity.

We have modified the manuscript accordingly, and detailed corrections are listed below point by point:

Response to Reviewer #1

This paper reports a comparative genomic analysis of the SAUR gene family among 'Dangshansu pear' (Pyrus 92 bretschneideri), Arabidopsis thaliana and Oryza sativa, and the role of PbrSAUR13 and PbrSAUR52 in lignin synthesis. This work will be helpful for the researcher in understanding the molecular mechanism of lignin biosynthesis and the content of stone cells of pears. However, I have several points that might need the author's attention.

  1. The first paragraph in the result section described the isoelectric points of SAUR proteins. However, I did not see the relevance of the isoelectric to this study as the author did not discuss it in either discussion or introduction section.

Reply: Thank you for highlighting the above problem and for your suggestions. We have checked the whole manuscript and added the isoelectric point. Kindly see in the revised manuscript.

  1. In the result section 2.1, SAUR genes number were reported least and most in groups 3 and 1 of the phylogenetic tree, respectively. Is there any evolutionary explanation for this? Please explain in the result or discussion section.

Reply: Thank you so much for your critical review. Now according to your suggestion, we have explained in the section 2.1. Kindly see the revised manuscript.

  1. Different Cis-acting elements were identified among SAUR genes, but no validation or confirmation has been provided about the roles of these elements and their impact on the gene expression. Does the author predict any connection between Auxin, PbrSAUR13, and 52, lignin biosynthesis genes?

Reply: Thank you so much for your critical review. It has been reported that auxin can promote the activity of cambium-specific and lead to the synthesis of lignin. The PbrSAUR gene family is an early response gene family of auxin, so we did a related experiment on the effect of overexpressing the PbrSAUR gene on the lignin content. We already mentioned in manuscript. kindly check revised manuscript.

Reference:

Zhang, L.; Kamitakahara, H.; Sasaki, R.; Oikawa, A.; Saito, K.; Murayama, H.; Ohsako, T.; Itai, A., Effect of exogenous GA4 + 7 and BA + CPPU treatments on fruit lignin and primary metabolites in Japanese pear “Gold Nijisseiki”. Scientia Horticulturae 2020, 272.

  1. Please explain whether PbrSAUR13 and PbrSAUR52 impact the number of stone cells or lignin contents only in the stone cells; it is unclear to me.

Reply: Thank you so much for your critical review.PbrSAUR13 and PbrSAUR52  affects the formation of stone cells by affecting the synthesis of lignin. Therefore, these two genes impact the number of stone cells and lignin contents.

  1. What was the data source to investigate the expression level of SAUR genes at different developmental stages? Please explain in the result/material and method section.

Reply:  Thank you very much for hardworking in careful review and valuable comments. We  already mentioned data source in section 4.7. kindly check in material and method.

  1. Each sample was replicated three times, but the Material and Method section did not mention how many samples were used.

Reply: Thank you so much for your critical review. The 39 DAF 'Dangshansu pear' fruit and the ginkgo stage Frandi strawberry were selected for further expression analysis. Now according to your suggestion, we have explained in the section 4.7 and 4.9.

  1. Please elaborate on the " excellent growth conditions" mentioned in the manuscript in multiple places. 

Reply: Many thanks for noticing this mistake. We apologize that this error. According to your suggestions, we updated and replace these words in full description. Kindly see in the revised manuscript.

Reviewer 2 Report

Dear Authors,

The manuscript is interesting and well written. I have some minor comments that may be helpful.

First of all, you have to highlight the purpose of the work. You have to highlight the purpose of the work.

Line 44: "9x103~3x104" what does it mean? Units are also missing.

Line 131: MeJA - It is necessary to explain what this abbreviation means.

In general, resolution of charts is low, so analyzing them is a challenge. Fig1-3 should be also larger. In Fig.3 cis-elements should be full filled instead of gradient, so that the graph become clearer. Fig 4 is redundant. Fig 5. The gaps between chromosomes should be smaller. Chromosomes should have various colors. Fig 7 should be moved to the supplement. Fig 8 - 10 do not have a marked scale. 

Latin names should be in italics and contain authority. 

In section, 4.9 it is not clear why you used strawberry?

Best,

M.

Author Response

Title: Comparative Genomic Analysis of SAUR Gene Family, Cloning and Functional Characterization of two genes (PbrSAUR13 and PbrSAUR52) in Pyrus bretschneideri

Dear Editor and Reviewers,

We thank all the Reviewers and the Editor for their careful reading of our manuscript and overall supportive comments and suggestions. This constructive input has contributed to improve the manuscript significantly, where we have also had a special eye to enhance clarity.

We have modified the manuscript accordingly, and detailed corrections are listed below point by point:

Response to Reviewer #2

  1. The manuscript is interesting and well written. I have some minor comments that may be helpful.

Reply: Thank you so much for kind remark for our manuscript.

  1. First of all, you have to highlight the purpose of the work. You have to highlight the purpose of the work.

Reply: Thank you so much for kind remark for our manuscript. I have re-emphasized the research purpose in the line 78.

  1. Line 44: "9x103~3x104" what does it mean? Units are also missing.

Reply: Thank you so much for kind remark for our manuscript. This is due to my spelling error; the correct representation should be 9 × 103~3 × 104. This data represents the relative molecular mass of the SAUR protein. The unit of relative molecular mass is 1

  1. Line 131: MeJA - It is necessary to explain what this abbreviation means.

Reply: Thanks for your valuable comment and we have modified and written full names of all abbreviations carefully in the revised manuscript. The revised version of the manuscript is more precise according to your suggestion.

  1. In general, resolution of charts is low, so analyzing them is a challenge. Fig1-3 should be also larger. In Fig.3 cis-elements should be full filled instead of gradient, so that the graph become clearer. Fig 4 is redundant. Fig 5. The gaps between chromosomes should be smaller. Chromosomes should have various colors. Fig 7 should be moved to the supplement. Fig 8 - 10 do not have a marked scale. 

Reply: Thanks for such important advice and suggestion on our manuscript. According to your good suggestion, we have modified all figures in our manuscript. Fig 8 - 10 scales have been marked in the lower left corner of the figure. Kindly check the revised manuscript. now this is word form due to resolution problem. We have rechecked each figure resolution.   we rechecked the resolution of the figure and again updated each figure. pixels/inch. Figure resolution on zoom also has very good quality. Kindly see the high-resolution figures by using below link.

https://1drv.ms/u/s!AqJXAGd8LreDmAsJ2nUsT7jDmQd8?e=gWDIlR.

  1. Latin names should be in italics and contain authority. 

Reply: Thanks for such important advice and suggestion on our manuscript. According to your good suggestion, we have modified all Latin names in our manuscript.

  1. In section, 4.9 it is not clear why you used strawberry?

Reply: Thanks for such an important and valuable question on our manuscript. Strawberry belongs to the Rosaceae species, and the pulp is white. If the lignin content changes, the observation will be more obvious, and strawberry is often used as a model crop of woody plants to determine the change in lignin content. In order, to further verify the function of PbrSAUR13 and PbrSAUR52, we then measured the lignin content of overexpressed strawberry fruit. kindly see the revised manuscript for more details.